# Color-tunable single-fluorophore supramolecular system with assembly-encoded emission

Qian Wang[1,4], Qi Zhang[1,4], Qi-Wei Zhang [2], Xin Li[3], Cai-Xin Zhao[1], Tian-Yi Xu[1], Da-Hui Qu [1*] & He Tian [1]

Regulating the fluorescent properties of organic small molecules in a controlled and dynamic manner has been a fundamental research goal. Although several strategies have been exploited, realizing multi-color molecular emission from a single fluorophore remains challenging. Herein, we demonstrate an emissive system by combining pyrene fluorophore and acylhydrazone units, which can generate multi-color switchable fluorescent emissions at different assembled states. Two kinds of supramolecular tools, amphiphilic self-assembly and γ-cyclodextrin mediated host-guest recognition, are used to manipulate the intermolecular aromatic stacking distances, resulting in the tunable fluorescent emission ranging from blue to yellow, including a pure white-light emission. Moreover, an external chemical signal, amylase, is introduced to control the assembly states of the system on a time scale, generating a distinct dynamic emission system. The dynamic properties of this multi-color fluorescent system can be also enabled in a hydrogel network, exhibiting a promising potential for intelligent fluorescent materials.

---

[1] Key Laboratory for Advanced Materials and Joint International Research Laboratory of Precision Chemistry and Molecular Engineering, Feringa Nobel Prize Scientist Joint Research Center, School of Chemistry and Molecular Engineering, East China University of Science and Technology, 130 Meilong Road, 200237 Shanghai, China. [2] School of Chemistry and Molecular Engineering, East China Normal University, Dongchuan Road 500, 200241 Shanghai, China. [3] Department of Theoretical Chemistry and Biology, School of Engineering Sciences in Chemistry, Biotechnology and Health, KTH Royal Institute of Technology, SE-106 91, Stockholm, Sweden. [4]These authors contributed equally: Qian Wang, Qi Zhang. *email: dahui_qu@ecust.edu.cn

Controlling molecular emission is a key topic for chemists, as it applies to many applications such as fluorescent imaging[1,2], light-emitting diodes[3], sensors[4–6], and photo-electric devices[7]. Exploring fluorophores that integrate high intensities[8–10], wavelength tunability[11,12], a wide concentration range[13], water solubility[14,15], and synthetic simplicity[16] is of broad interest. Recently, molecular emissive systems, which exhibit intelligence, have been brought into focus because of the emerging interest in smart molecular materials that respond to external stimuli[17–20]. Such smart fluorescent materials are expected to produce emission with the desired intensity and wavelength under external interferences. This intelligence can endow fluorescent dyes versatility in many potential applications, such as encryption materials[21–23], super-resolution microscopy[24], and controllable imaging probes[25].

Traditional strategies to control molecular emission mainly involved chemical covalent modifications of the fluorophores to shift the energy levels or generate energy/electron transfer processes[26,27]. However, the disadvantages of this strategy include the tedious organic synthesis, static molecular structures, and nonadjustable emission properties, of which the last is quite unfavorable for smart fluorescent materials. Recently, some ingenious strategies have been exploited to dynamically modulate fluorescent emission[28–30]. Tang et al. have developed a series of fluorophores whose aggregates emit more strongly than monomers, called aggregation-induced emission (AIE)[31–33]. The AIE fluorophores have proven to be effective as turn-on fluorescent sensors, thus exhibiting great potential in the biological imaging field[34]. Alternatively, supramolecular chemists have discovered fluorophores whose emission properties can be affected by noncovalent interactions[14,16,35,36]. The amphiphilic assemblies have been proven to be reliable to support efficient energy transfer among multiple fluorophores[20]. De Cola and co-workers exploited an amphiphilic self-assembly strategy to control the emission wavelengths of a single fluorophore with a varied self-assembly pathway at different time scales[37]. Our group also demonstrated that the emission wavelengths could be varied to produce multi-color emission in a single supramolecular system[35,36]. However, it still remains unsolved whether one can artificially control the molecular emissive system in a biomimetic mode, in which the emission properties, especially the emissive wavelength, can be dynamically modulated in a highly controllable way.

To that end, we herein demonstrate that the emission color of a single fluorophore can be dynamically controlled and modulated in the blue-to-yellow region, including white-light emission. The two different controlling modes involve intrinsic amphiphilic self-assembly and molecular recognition with external chemical signals and a water-soluble macrocycle, γ-CD[38,39], respectively. Taking advantage of the distinct excimer emission features of the pyrene moiety, we reveal the strong relationship between the visualized emission color and the complex molecular self-assembly, enabling a single fluorophore to produce variable multi-color fluorescent emission, which can be dynamically manipulated by localized concentrations and chemical signaling molecules. Meanwhile, we also expand this unique assembly-encoded emission system toward a biomimetic enzyme-catalyzed hydrolysis[40] reaction as well as the smart hydrogel-based fluorescent materials.

## Results

**Molecular design.** Pyrene units are widely used as a fluorophore, which produces strong red-shifted excimer emission when stacked together in a dimer state[41–43]. This feature has enabled the successful construction of many stimuli-responsive fluorescent sensors. However, previous examples only covered very limited emission regions, in which the emission wavelength ($\lambda_m$) did not exceed 500 nm[41,44]. The spectrum window produced by stacking-induced excimer emission was too narrow to enable complex controlling using multi-colored fluorescence as the signal output. To broaden this window toward the red-shifted region, we introduced an acylhydrazone unit into a pyrene moiety[45] (Fig. 1). There are two reasons to perform this modification: (i) the introduced fluorochromes and slightly enhanced π-conjugated backbone might result in a red-shift of the fluorescence; and (ii) the potential intermolecular hydrogen bonds in

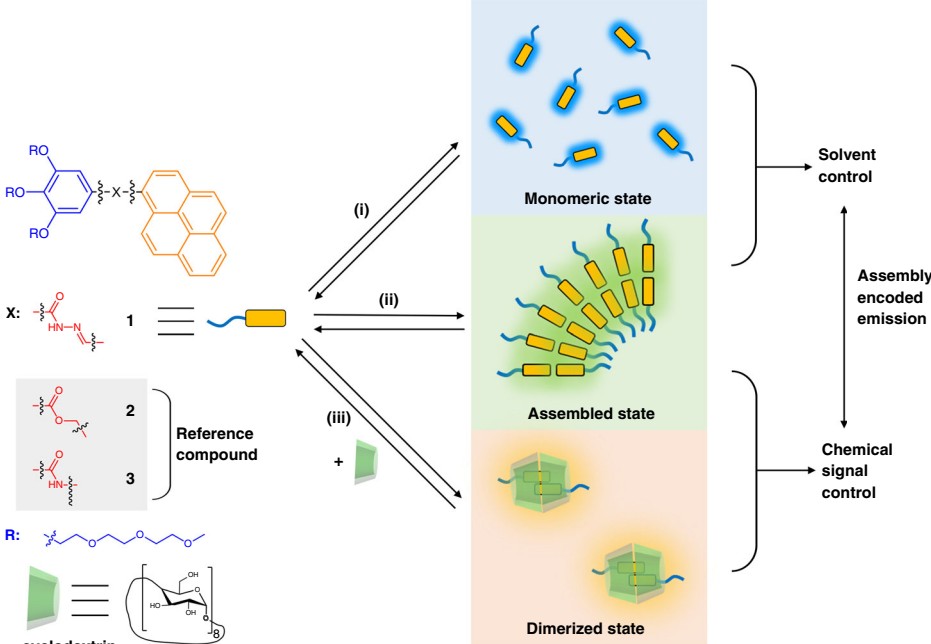

**Fig. 1 Molecular structures and schematic representation of multi-color emissive systems. (i)** monomeric state in organic solvents; **(ii)** amphiphilic self-assembly in aqueous media; **(iii)** dimerization in the cavity of host macrocycles.

the acylhydrazone units may work as secondary noncovalent interactions to stabilize the primary stacking interactions and make subtle differences in the stacking parameters of the pyrene units. Meanwhile, the other terminal side of the acylhydrazone pyrene was covalently modified with three hydrophilic glycol chains, making the whole molecule a typical amphiphilic model. This feature could enable amphiphilic self-assembly in aqueous media, driving the hydrophobic pyrene units to stack as an inner membrane moiety, and thus turning on excimer emission. Compound 1 was synthesized in an accessible route by sequential esterification, hydrazinolysis, and acylhydrazone formation. Meanwhile, the control molecules, compounds 2 and 3, were also synthesized for comparison, which bear a similar amphiphilic structure with the ester (compound 2) and amide units (compound 3) instead of an acylhydrazone unit. The three molecular structures were confirmed by NMR and mass spectroscopy (Supplementary Figs. 1–12).

**Amphiphilic self-assembly.** The amphiphilic feature of compound 1 enables its self-assembly in aqueous solutions (Fig. 2a). Dynamic light scattering (DLS) measurements revealed that the particle size of compound 1 aqueous solution was around 100 nm (Fig. 2b). Morphologies of the aqueous assemblies were detected by transmission electron microscopy (TEM) (Fig. 2c, Supplementary Fig. 13) showing hollow spheres with a diameter of about 110 nm and a membrane thickness of $2.6 \pm 0.5$ nm. This result suggested the vesicle assemblies of compound 1 in water. Then the concentration-dependent UV–Vis absorption spectra of

compound 1 in aqueous solutions (from 5 to 13.5 μM) showed a considerable red-shift ($\Delta\lambda_m = 24$ nm) and an enhancement of the molar extinction coefficient ($\varepsilon$) at around 400 nm (Fig. 2d). Three distinctive isosbestic points at 290, 325, and 350 nm suggested the occurrence of induced aromatic stacking by amphiphilic self-assembly. A distinctive slope transition point in the linear function of $\varepsilon_{380nm}$ versus concentration revealed a critical aggregation concentration (CAC) of 9 μM (Fig. 2d inset). In organic solvents, sharp absorption bands at 400 nm were observed at 30 μM (Supplementary Fig. 14), indicating the non-aggregated status of compound 1 in organic solvents.

The fluorescent emission of compound 1 was then investigated. A sharp blue emission peak at 420 nm was observed in $CH_2Cl_2$ solution, which was a distinctive emission attributed to the monomeric pyrene units (Supplementary Fig. 15). Increasing the concentration of compound 1 in $CH_2Cl_2$ solution made little difference in the emissive peak, indicating the non-assembled nature of compound 1 in organic solvents. In aqueous solution, a diluted concentration (5 μM) of compound 1 resulted in a blue emission at 460 nm, while a broad emission peak at 530 nm, with a large Stokes shift (120 nm), rose remarkably with increased concentrations (Fig. 2e). The plot of the fluorescence intensity at 530 nm versus the concentration revealed the CAC was 9 μM (Fig. 2e inset), which was consistent with the result measured by the concentration-dependent UV–Vis spectra (Fig. 2d inset). The observed emission peak at 530 nm was attributed to the excimer emission of the stacked pyrene units induced by amphiphilic self-assembly and intermolecular hydrogen bonds (Supplementary

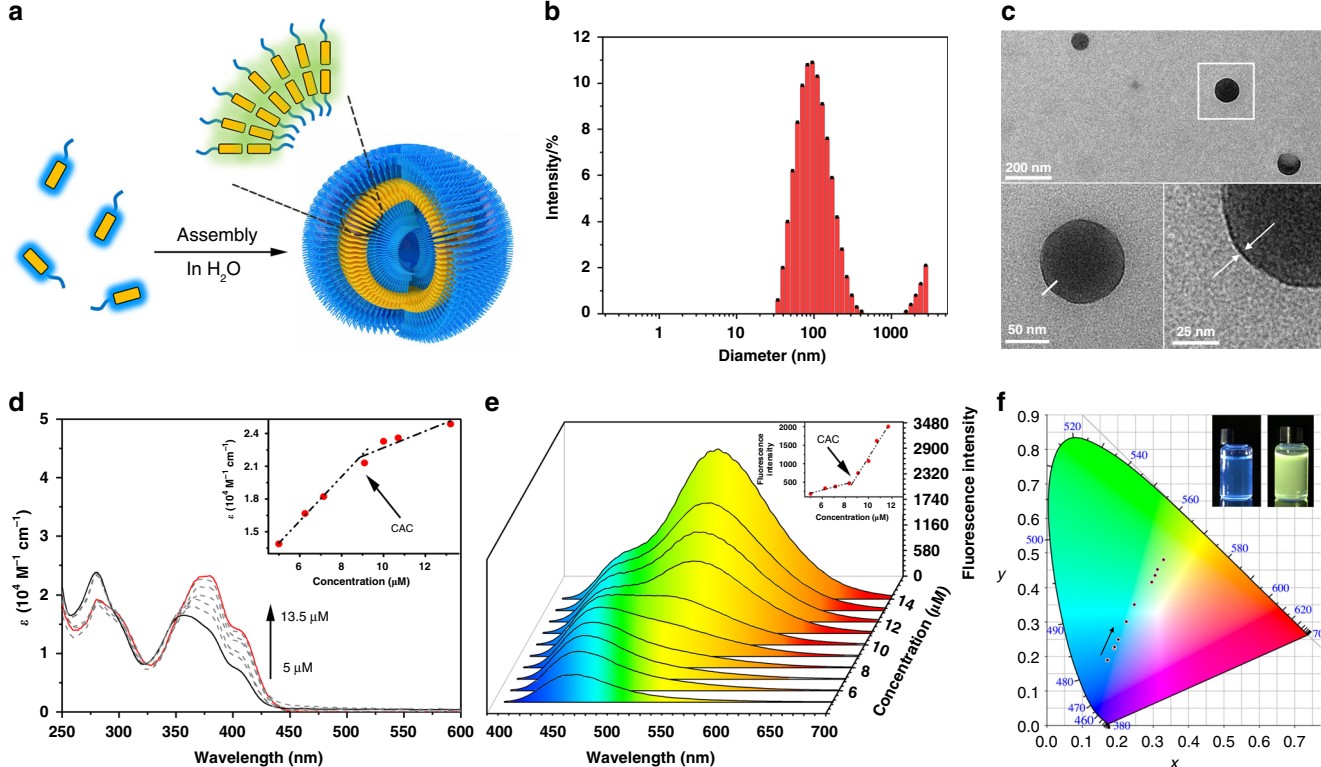

**Fig. 2 The amphiphilic self-assembly and photophysical studies of compound 1. a** Schematic representation of the self-assembly behavior of compound 1 in an aqueous solution. **b** DLS data of compound 1 in an aqueous solution (25 °C, 0.1 mM). **c** TEM images of compound 1 that self-assembles to form vesicles, in an aqueous solution (25 °C, 0.1 mM). **d** Concentration-varied UV–Vis spectra of compound 1 in aqueous solutions from 5 μM (black solid line) to 13.5 μM (red solid line), indicating self-assembly of compound 1. The inset shows a plot of the extinction coefficient $\varepsilon$ versus the concentration, revealing the critical aggregation concentration. **e** Concentration-varied fluorescence spectra of compound 1 in aqueous solutions (from 5 to 13.5 μM), $\lambda_{ex} = 365$ nm. Inset: the plot of the fluorescence intensity at 530 nm versus the concentration, revealing the critical aggregation concentration as 9 μM. **f** CIE 1931 chromaticity diagram of compound 1 with increasing concentration (from 5 to 13.5 μM). Inset shows the visible fluorescence of compound 1 (left: 5 μM, right: 13.5 μM) under the exposure of a 365 nm light.

Fig. 16). To confirm the formation of the excimer, the excitation spectra were measured (Supplementary Fig. 17). Excitation of the low concentration solution (5 μM, monomer) resulted in a similar emission as that of the concentrated solution (20 μM, aggregation), suggesting the presence of the excimer. The color coordinates of the fluorescence were plotted in a CIE chromaticity diagram (Fig. 2f), showing a color variation from blue (0.18, 0.18) to green-yellow (0.32, 0.42) at different concentrations of aqueous solutions. To investigate the solvent effects on the fluorescent properties, fluorescence spectra of compound 1 in various solvents were measured (Supplementary Figs. 18 and 19). Notably, a blue-white emission (0.26, 0.35) could be obtained in DMSO/$H_2O$ mixtures with high water fractions (Supplementary Fig. 20), showing a single-compound white-light emission system.

To understand the role of the introduced acylhydrazone unit, two reference compounds 2 and 3 were designed to replace the acylhydrazone unit into ester bond and amide bond (Fig. 1). The amphiphilic self-assembly of compound 2 was investigated by TEM, UV–Vis, and fluorescence spectra (Supplementary Figs. 21 and 22). The sharp absorption bands at 315, 325, 340 nm in $CH_3OH$ (20 μM) as well as $H_2O$ (1 μM) suggested the existence of the free monomer while the concentrated aqueous solution showed a red-shift of 10 nm of the broad absorption bands at around 340 nm, suggesting the aggregation state. In addition, an excimer emission with a wavelength at 490 nm was observed in the concentrated aqueous solutions of compound 2. Similar optical properties were observed in compound 3 solutions (Supplementary Fig. 23). The substantial difference in the aggregation–emission relationship among the three amphiphilic compounds was attributed to the key role of the acylhydrazone units in compound 1. Therefore, the acylhydrazone units substantially enhanced the stacking interactions of the π-conjugated pyrene backbone, resulting in a red-shift of the fluorescence and generating the green-yellow emission.

**Host–guest molecular recognition**. According to the above demonstration, compound 1 can perform amphiphilic self-assembly and generate strong red-shifted excimer emission owing to the assembly-induced stacking of the pyrene units. Next, it was necessary to determine whether the excimer emission could be further red-shifted by more closely confining the pyrene fluorophores. Macrocycles bear large cavities with specific chemical environments to selectively bind guest molecules inside. For example, CD is a species of macrocycle with hydrophobic cavities. Considering the excellent host–guest combination between the CD macrocycle and a hydrophobic molecule, we further explored whether it was possible to bind the pyrene unit in compound 1 as a closely stacked dimer in the cavity of a CD macrocycle to determine if this could result in distinct emission properties.

A UV–Vis titration experiment was performed by adding γ-CD into the aqueous solution of compound 1 (Fig. 3a). A slight decrease of the molar extinction coefficient ($ε$) was observed at around 380 nm when the amount of γ-CD was <2 eq, while $ε_{380}$ remarkably rose after the addition of 2–10 eq of γ-CD. The nonlinear correlation between $ε$ of the guest, compound 1, and the concentration of the host, γ-CD, suggested a possible multistep host–guest interaction in this system. Particularly, the sharp extinction peak at 380 nm indicated the formation of pyrene dimers. For comparison, α-CD and β-CD macrocycles with smaller cavities were also used to titrate the compound 1 aqueous solution, showing no distinctive peak changes (Supplementary Fig. 24). Considering the cavity of the γ-CD macrocycle is chiral, one way to determine whether guest chromophores were recognized was to look for the strong circular dichroism signal produced by delivery of the spatial chirality. Hence, circular

dichroism spectra were tested, which showed a strong Cotton effect at 380 nm (corresponding to the sharp extinction peak in the UV–Vis spectra) for the mixed aqueous solution of compound 1 and γ-CD (Fig. 3b), indicating that the pyrene units were included in the chiral cavity of the γ-CD as a dimer. No similar distinctive signal was observed in the samples of α-CD and β-CD macrocycles.

Isothermal titration calorimetry (ITC) measurements further indicated that the molar ratio of the host–guest system was 1:1 (Fig. 3c). Combining the evidence of the pyrene dimer revealed by the UV–Vis titration and CD spectra, we concluded that the binding mode of the host–guest system between compound 1 and γ-CD was the 2:2 type instead of the 1:1 type. The binding mode was further confirmed by NMR spectroscopy and mass spectrum (Supplementary Figs. 25–27). Notably, the binding constant $K_a$ was measured as $(6.11 ± 0.48) × 10^5 \, M^{-1}$, which was an unusually high affinity for γ-CD-based host–guest systems. This can be explained as the close aromatic stacking in this distinct 2:2-type binding mode as well as possible secondary hydrogen bonding interactions between the hydroxy groups of the γ-CD and the acylhydrazone units of compound 1.

To better understand the structure of the host–guest system, the geometries of compound 1 (monomer and dimer in the absence and presence of γ-CD) were optimized using the MMFF94 force field[46] as implemented in the Avogadro software[47]. At the optimized geometries, frontier molecular orbitals were calculated using density functional theory with the hybrid PBE0 functional[48] and the 6-31+G(d) basis set[49], as implemented in the Gaussian 09 program package. The simulated HOMO and LUMO energy levels were calculated (Fig. 4a, Supplementary Fig. 28). The HOMO–LUMO gap decreased with the formation of the stacking conformation, which corresponds to what was observed in the fluorescence spectra.

The fluorescence spectra of compound 1 titrated with the γ-CD macrocycle (Fig. 4b) showed a remarkable change. With the concentration of the γ-CDs increased, the monomer fluorescence of compound 1 at 460 nm gradually decreased, while an excimer emission rose at 560 nm (Supplementary Fig. 29). The Stokes shift, as large as 185 nm, was attributed to the formation of a closer stacking conformation of the pyrene moieties in the confined cavity, which narrowed the energy gap between the HOMO and LUMO. Notably, owing to the dual-color emission produced by the monomer/assembly mixed system, white-light emission could be observed by the simple addition of the γ-CDs to a dilute solution of compound 1 or by immersing a fluorescent gel into the γ-CDs aqueous solution. The fluorescent color coordinates were calculated and plotted in a CIE 1931 chromaticity diagram (Fig. 4c). The coordinates changed from blue (0.18, 0.18) to near white (0.29, 0.33), (0.31, 0.35) and yellow (0.37, 0.43) with increasing concentrations of γ-CD (0–10 eq). The multi-color fluorescent emission of the mixture of 1 and γ-CD with different molar ratios was visibly present in aqueous solution and hydrogen gel (Fig. 4d, e, respectively). Moreover, the Stokes shift observed with compound 1 was larger than with both compounds 2 and 3 in the presence of γ-CD (Supplementary Figs. 30 and 31). No increase in the long wavelength emission was detected upon the addition of excess α-CD or β-CD (Supplementary Fig. 32). The dilute aqueous solutions of compound 1 were weakly fluorescent ($Φ = 5.65\%$), whereas the emission of the vesicles was enhanced ($Φ = 36.36\%$), and the emission induced by the assembly of the pyrene moieties and the γ-cyclodextrins was increased by one order of magnitude ($Φ = 46.21\%$, related to dilute solution; Fig. 4f). In addition, the measurement of absolute solid fluorescence quantum yield was performed (Supplementary Fig. 33). The quantum yield of compound 1 was $Φ = 17.05\%$, whereas the quantum yield of the complex increased up to

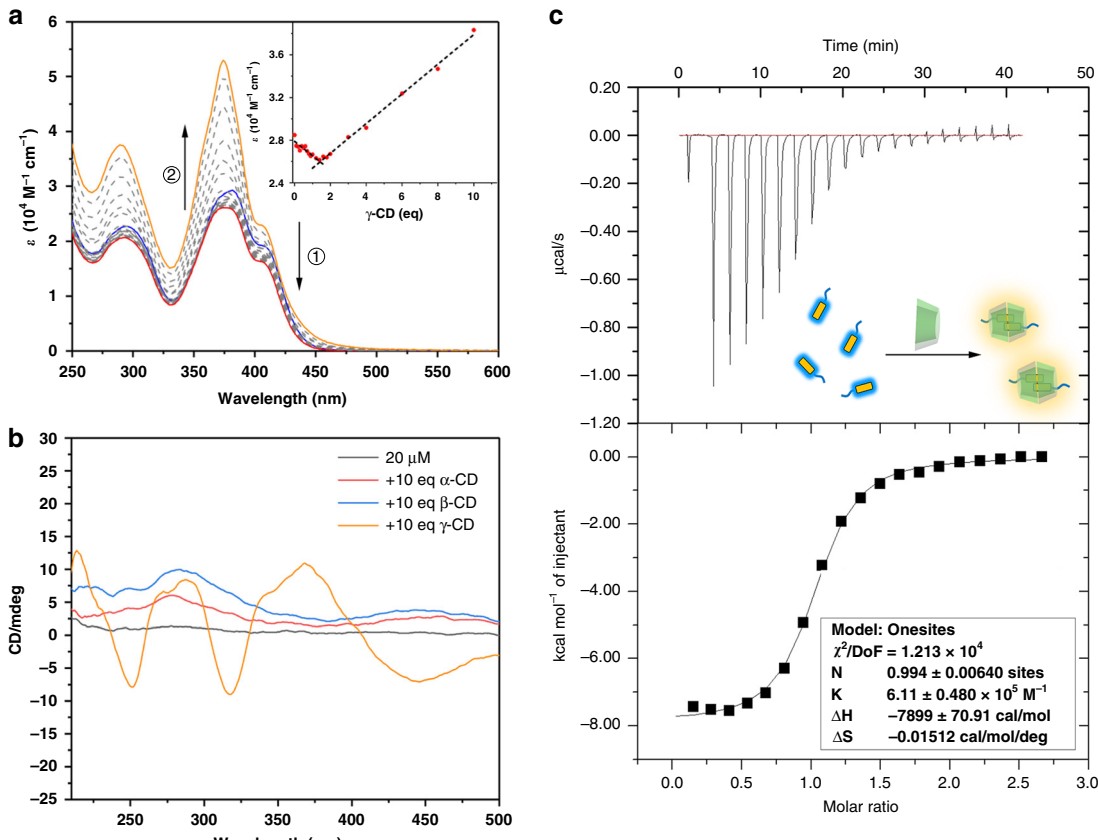

**Fig. 3 Host–guest recognition with CD macrocycles. a** Changes in the UV–Vis absorbance of compound 1 in $H_2O$ (25 µM) upon titration with a γ-CD aqueous solution (10 mM), blue line: spectra before γ-CD addition; red line: spectra after 1.5 equivalents of γ-CD were added; orange line: absorption after 10 equivalents of γ-CD were added. Inset: extinction coefficient changes of compound 1 at 374 nm as a function of the molar ratio of γ-CD. **b** CD spectra changes of compound 1 before and after the addition of the three types of CD macrocycles. **c** ITC data for compound 1 with γ-CD. Buffer solution: PBS = 0.05 M (pH = 5.2), cell: γ-CD = 0.07 mM, syringe: compound 1 = 0.75 mM.

$Φ'$ = 47.99%, owing to the assembly. The fluorescence lifetime of the complex was longer than the monomer as well as the vesicles. This may be attributed to the formation of the dimer in γ-CD (Supplementary Fig. 34). The multicolor photoluminescence materials with bright fluorescence may be a potential candidate for message storage and transduction[21–23]. Message storage in the aqueous solution was presented (Supplementary Fig. 35) using a 96-well plate as a template. Furthermore, the fluorescent aqueous solution could also be used as a printing ink for fluorescent patterning on cellulose paper (Fig. 4g, Supplementary Fig. 36), showing its versatility and processability as a typical fluorescent material.

**Dissipative molecular emission system**. The multi-color fluorescent property, which is highly dependent on the concentration of the chemical signal γ-CD, leads us to an intriguing hypothesis: can we design the thermodynamically controlled fluorescent system on a time scale and push the system out of equilibrium[50–54]? Our strategy was to introduce an enzyme-catalyzed hydrolysis reaction to be integrated with the supramolecular self-assembly process. The control mechanism can be demonstrated as following (Fig. 5a): The chemical signal γ-CD and α-amylase were added to an aqueous solution of compound 1. The quaternary supramolecular assemblies were formed immediately, leading to the accompanying fluorescent color change from blue to yellow. Then, the α-amylase would trigger the slow hydrolysis reaction to decompose the macrocycle γ-CD into non-macrocyclic

oligosaccharide products, thus driving the disassembly of the yellow-emissive quaternary supramolecular assemblies into blue-emissive monomers. This process could enable an out-of-equilibrium fluorescent system in which the fluorescent color jumps from blue to yellow and then automatically and slowly reverts from yellow to blue.

In a typical experiment, the mixed solution displayed a blue monomer emission at 460 nm (Fig. 5b), indicating that the fluorescence did not interfere with the enzyme and phosphate buffer (UV–Vis spectra in Supplementary Fig. 37). Upon addition of the γ-CDs to the system, efficient excimer emission at 560 nm increased owing to the spontaneous host–guest recognition. Then, such emission gradually decreased with the slow hydrolysis of the γ-CDs. After 2 h, the cyclodextrins completely decomposed and a solution with blue emission was obtained, indicating the complete disassembly of the pyrene moieties. In a control experiment, the fluorescent color was unchanged, even after 72 h in the absence of γ-CDs (Supplementary Fig. 38). The chemically fueled out-of-equilibrium system could be driven over three cycles (Supplementary Fig. 39). Additionally, the enzymatic reaction of the γ-CDs was found to be a self-regulated system, i.e., the products (maltooligomers, glucose, maltose) of the enzymatic reaction worked as feedback for the hydrolysis rate of the γ-CDs[55], resulting in increased lifetimes after each cycle (Fig. 5c, Supplementary Fig. 39). The change of the fluorescent color is presented in a CIE chromaticity coordinate (Fig. 5d). By adding γ-CDs to the solution, the luminescence color jumped from blue (0.16, 0.18) to yellow (0.35, 0.39). Then, the color

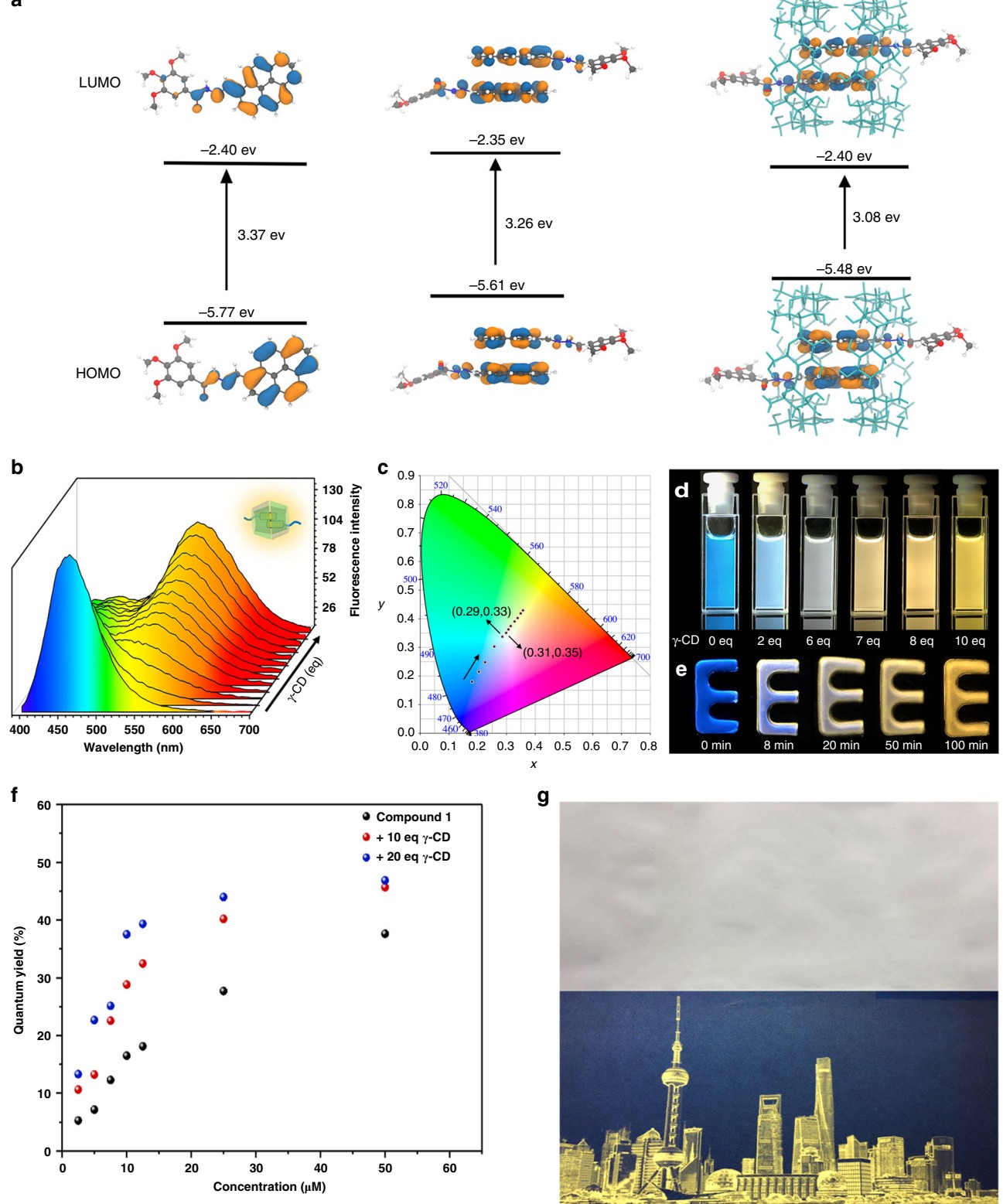

**Fig. 4 Multi-color emissive properties of the host–guest systems. a** The calculated HOMO and LUMO energy levels of compound 1. **b** Changes in the fluorescence spectra upon the addition of γ-CD (compound 1: 5 μM, γ-CD: 0–10 eq). **c** Changes in the 1931 CIE chromaticity coordinate with the addition 0–10 eq of the γ-CD host to an aqueous solution of compound 1 (5 μM). **d** Fluorescence images of a solution of compound 1 (5 μM) upon addition of increasing concentrations of γ-CD aqueous solution (0−10 eq). **e** Fluorescence gel immersed in γ-CD aqueous solution (0.1 M) for 8, 20, 50, and 100 min. **f** Absolute fluorescence quantum yield of compound 1 at different concentrations upon the addition of γ-CD. **g** Fluorescent photographs printed by inkjet using a black cartridge that was loaded with the complex solution under natural light and UV.

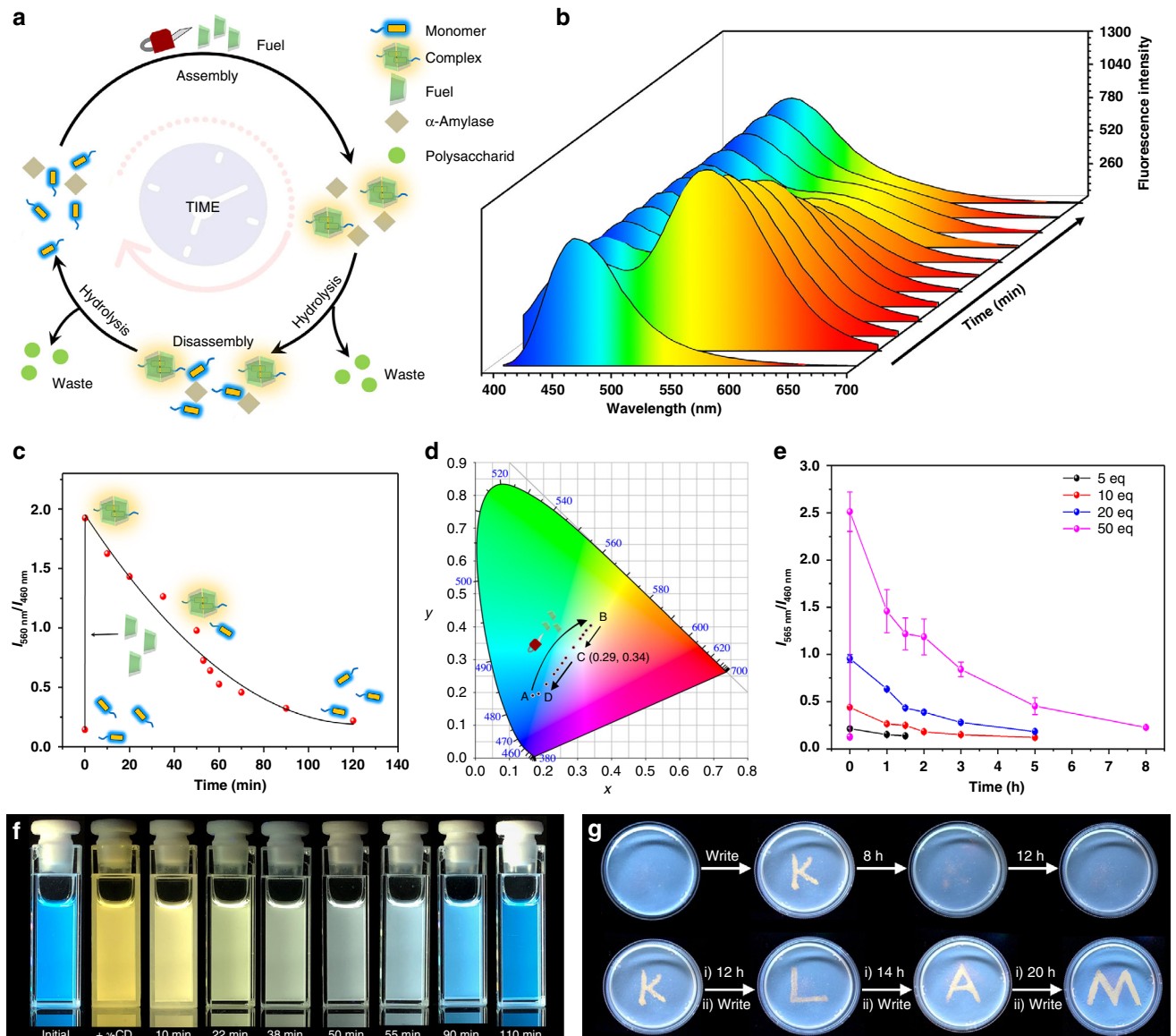

**Fig. 5 Controlling the multi-color fluorescent system on a time scale. a** Schematic representation of the transient assembly cycle of compound 1 and γ-CD, resulting in an out-of-equilibrium multicolor fluorescence. **b** Time-dependent fluorescence spectra showing the change of the fluorescence (from yellow to blue; compound 1 3 μM, α-amylase 1 mg ml$^{-1}$, phosphate buffer pH = 5.2). **c** Time-dependent relative fluorescence intensity. The delay rate of the relative fluorescence intensity gradually decreased, suggesting a self-regulated mechanism. **d** Time-dependent CIE coordinate diagram corresponding to **b**, showing the trajectory of the out-of-equilibrium emission color change. **e** The concentration of the chemical fuel regulated by timescale (compound 1 5 μM, γ-CD 5−50 eq, α-amylase 1 mg ml$^{-1}$, phosphate buffer pH = 5.2). Data are presented as the mean ± s.d. (n = 3). **f** A series of fluorescence images of the out-of-equilibrium fluorescence. **g** Writing into a self-erasable fluorescence gel (compound 1 1 mM, α-amylase 1 mg ml$^{-1}$, phosphate buffer pH = 5.2). (i) Messages stayed for the specific hours. (ii) Chemical fuel input by writing with γ-CD aqueous solution.

autonomously turned to white (0.29, 0.34) and ultimately reverted back to the initial blue color (0.18, 0.18) with time. The lifetime of the out-of-equilibrium supramolecular system can be turned by controlling the concentration of the fuel (Fig. 5e). For instance, it can be changed from a short-lived (90 min), weak fluorescence at γ-CDs = 25 μM (5 eq) to a brighter fluorescence with lifetimes of 480 min at γ-CDs = 250 μM (50 eq). Remarkably, upon the addition of the γ-CDs, the solution displayed a yellow luminescence, which then reverted to a yellow-white-blue color (Fig. 5f), which corresponds to the CIE chromaticity coordinate.

The application of self-erasing materials in secure message storage and communications has attracted a lot of attention in recent years. So far, several self-erasing materials have been successfully developed by using various inks[56,57]. A multicolor

photoluminescence material that has a transient fluorescence with a tunable lifetime may be a potential candidate for generating self-erasing materials. However, transferring dissipative fluorescence from the solution phase to the solid phase, which is more efficient for fabricating devices, is a challenge. To achieve this goal, a functional paper, a blue-emission hydrogen-gel, was prepared by thermal-initiated gelation (Fig. 5g). The aggregation of compound 1 disassembled in PEG solution before gelation, owing to decreased hydrophobic associations and destruction of hydrogen bonds. Thus, the hydrogen-gel displayed a blue emission. Messages can be written with a paintbrush with a γ-CDs aqueous solution. The written message was displayed with a bright yellow luminescence, which was in strong contrast to the luminescence color of the paper, due to the supramolecular

assembly in the active environment. The energy then dissipated through two methods (i) it diffused into the hydrogen gel or (ii) hydrolysis by α-amylase. Thus, the active environment became dormant, i.e., the complexes disassembled, and the messages gradually self-erased. The written-erased process could be repeated over four times. Both the fluorescence intensity and the lifetime of the messages changed with a change in the concentration of the ink. A total of 50 mM of ink created a more active environment and a longer lifetime (12 h), while a weakly fluorescent message with a short lifetime (1 h) was observed when using 1 mM of ink (Supplementary Fig. 40). In addition, the lifetime of the materials could be turned with an enzyme. The paper with an enzyme (1 mg ml$^{-1}$) could be rewritten on over four times, whereas it could only be rewritten on twice without the enzyme (Supplementary Fig. 41).

## Discussion

In summary, we have achieved an out-of-equilibrium white-light emissive material with multicolor photoluminescence created by a coupled supramolecular assembly with a single fluorophore molecule and cyclodextrin that efficiently changes the fluorescence with a chemical reaction network. In addition, the fuel was used as a specific ink to develop a kind of self-erasing material, which was suitable for storing secretive and temporal information. Comparing to stimuli responsive materials, these materials are more closely related to autonomous and intelligent natural biomaterials, a key property for next-generation materials. We expect that this work will be a significant advancement for multicolor fluorescent materials and provide a strategy for constructing out-of-equilibrium systems.

## Methods

**Materials**. Chemicals were purchased from TCI, Adamas-beta® and Sigma-Aldrich and used without any further purification. All solvents were reagent grade and were dried and distilled prior to use according to standard procedures.

**Compound synthesis and purification**. The synthetic details of compounds 1, 2, and 3 can be found in supplementary Figs. 1, 5, 9, respectively. The molecular structures are determined using $^1$H NMR, $^{13}$C NMR spectroscopies, and high-resolution electronic spray ionization (ESI) mass spectrometry.

**Equipment**. Chemicals were weighed on analytical balances METTLER-TOLEDO, ME204T/02. $^1$H NMR and $^{13}$C NMR measurements were performed on a Brüker AV-400 spectrometer at room temperature. The ESI mass spectra were measured on an LCT Premier XE mass spectrometer. Ultraviolet–visible spectroscopic measurements spectra were acquired on a Varian Cary 100 spectrometer (1 cm quartz cells). The fluorescence spectra were performed on a Lengguang Luminescence spectrophotometer F97PRO. The morphologies were detected by transmission electron microscopy (JEM2000EX). DLS was measured on MALVERN, ZETA SIZER, ModelZEN3600 at 303 K.

**Ink printing tests**. Printing tests were performed on a Canon inkjet printer (MG 2400) and Canon PG-845 FINE cartridge, using paper without optical brightener. Ink in black cartridge was replaced by the aqueous solution consisting of compound 1 (1 mM) and γ-CDs (10 eq). Fluorescence images that printed on the paper was visible under UV light.

**Preparation of fluorescence gels**. The poly (ethylene glycol) methyl ether methacrylate (average Mn = 2000, 50 wt% in H$_2$O, Aldrich) and poly (ethylene glycol) diacrylate (average Mn = 700, Aldrich) were used as received. An aqueous solution consisting of phosphate buffer (pH = 5.2, 3 ml) PEG methyl ether methacrylate (2.55 g), poly PEG diacrylate (450 mg), enzyme (3 mg, control experiment: without enzyme) and compound 1 (1 mM) was prepared. K$_2$(SO$_4$)$_2$ (0.1 g) was added to the solution under stirring until dissolved. Then N, N, N, N-Tetramethylethylenediamine (10 μl) was added. The radical polymerization was then preceded for 5 min at 40 °C.

## Data availability

The data that support the findings of this study are available from the corresponding author upon request.

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

## Acknowledgements

This work was supported by National Natural Science Foundation of China (grants 21788102, 21790361, 21871084, 21672060, 21421004), Shanghai Municipal Science and Technology Major Project (Grant no. 2018SHZDZX03), the Fundamental Research Funds for the Central Universities, the Program of Introducing Talents of Discipline to Universities (Grant B16017), Program of Shanghai Academic/Technology Research Leader (19XD1421100), and the Shanghai Science and Technology Committee (Grant 17520750100). We thank the Research Center of Analysis and Test of East China University of Science and Technology for help on the material characterization.

## Author contributions

Q.Z. and D.-H.Q. conceived the project and designed the molecules. Q.W. performed the compound syntheses and characterizations. X.L. performed the DFT calculation. C.-X.Z. performed the TEM experiments. T.-Y.X. assisted in the design of figures. Q.-W.Z. discussed the experimental results and gave suggestions. Q.W., Q.Z., D.-H.Q., and H.T. wrote the manuscript. All authors discussed the results and commented on the manuscript.

## Competing interests

The authors declare no competing interests.
