## [Peer Review File · Nature Communications]

Reviewers' comments:

Reviewer #1 (Remarks to the Author):

In this manuscript, Qu et al. designed a single fluorophore based on amphiphilic pyrene by introducing an acylhydrazone unit to broaden pyrene pi-conjugated backbone and to facilitate the intermolecular hydrogen-bonding interactions. On one hand, this single fluorophore could self-assemble into spherical particles in aqueous solution due to its inherent amphiphilic structure, thus leading to the formation of the excimer. On the other hand, gamma-cyclodextrin was used to draw pyrene fluorophores much closer for the red-shifted excimer emission. Moreover, the chemical input of alpha-amylase was utilized to decompose the macrocyclic gamma-CD, thus resulting in the programmable excimer emission. Interestingly, this multi-color fluorescent system could also be applied as self-erasing/ink materials. The conceptual design and methodology are novel and the conclusions are supported by the experimental results. Overall, in my view, the present work is suitable for publication in Nature Communications after addressing the following concerns.

Comments:

1. In Scheme 1, the cartoon diagram of host macrocycle in green should be marked as gamma-cyclodextrin and the corresponding chemical structure should be given.
2. In the part of "Amphiphilic self-assembly", the reduplicate TEM images in Figure 1C and S14 should be removed.
3. In the part of "Amphiphilic self-assembly", is the morphology of the aqueous assemblies spherical vesicles or micelles? It cannot be well assigned as bilayer structure from the existing representation. Is there any stronger evidence to prove that they are vesicles? Otherwise, amphiphilic nanoparticle may be an appropriate expression for the self-assembling morphology.
4. The authors used the mixed solution of gamma-cyclodextrins and compound 1 to print the fluorescent images in Figures 3G and S37, but why was the color of the fluorescent patterning slightly different?
5. In the part of "Dissipative molecular emission system powered by chemical fuel", the fluorescent color was unchanged even after 72 h in the absence of gamma-CDs, the picture should correspond to Supplementary Figures S39, not S30.
6. Considering the importance of the 2:2 complex, more direct evidences were suggested to be provided to verify the formation of host-guest complex, such as 2D NOESY, and/or small-angle X-ray scattering (SAXS), and/or high-resolution mass spectrum, to comprehensively characterize the binding/aggregating mode between compound 1 and gamma-CD.
7. In Figure 1B, what does vertical coordinate (intensity/%) of DLS data refer to?
8. In the legend of Figure 1E, the value of CAC (90 μ M) is different from the description (9 μ M).
9. In Figure S13B, this reviewer cannot understand the large difference in length-width ratio. Why the height of spherical vesicle is 65 nm but the width is as large as 3 μ m?
10. In Page 10, last line, the K_a value should be changed to $(6.11 \pm 0.48) \times 10^5 \text{ M}^{-1}$.
11. In page 11, last line, the pH value and ionic strength of PBS buffer should be given.
12. Authors claimed that the products of the enzymatic reaction resulted in the increase of lifetime and explained as feedback. This reviewer is curious about what kind of product decreased the reaction rate and how it occurred? Some reasonable explanations should be supplied.

Reviewer #2 (Remarks to the Author):

The manuscript by Qu et al. reports the color-tunable single-fluorophore, which showed interesting color-responding emission via assembled molecules. By using pyrene fluorophore and the acylhydrazone linker, the formed supramolecular system in aqueous media, gives the emissions ranging from blue to yellow, even a pure white light emission, while also shows fluorescent color differences within a slow hydrolysis reaction. In principle, there are certain interesting results in this contribution to smart fluorescent materials. However, there are several misleading

descriptions and unclear presentations in the claim and the conclusion. This paper in the present form therefore could not be acceptable although significance and novelty of the content are high enough. A reconsider with major revisions should be recommended. Several important points to improve and strengthen the paper are listed below:

1. The authors should be careful when the word "programmable", do you think the emission is programmable? "precisely control the intermolecular aromatic stacking distances", how to 'precisely control'? these description and conclusion are overstated.
2. Page 6. ".....transmission electron microscopy (TEM) (Supplementary Figures S13 and 14), showing spherical vesicles" please provide the amplified image of the vesicles in TEM, and give the Rg/Rh value of TEM.
3. Page 7. "To confirm the formation of the excimer....., suggesting the presence of the excimer". Please describe the excimer more clearly and explain in detail, and then give the conclusion.
4. Page 7, ".....revealed a critical aggregation concentration (CAC) of 9 μM " and ".....and versus the concentration revealed the CAC was 9 μM ". While figure 1D and 1E which give 90 μM ?
5. Page 8, ".....could be obtained in DMSO/H₂O mixtures with high water fractions (Supplementary Figure S20)" the data is not provided in supporting information "Figure S20".
6. The authors claim "The potential intermolecular hydrogen bonds in the acylhydrazone units may work as secondary noncovalent interactions to.....". In this case, evidences of hydrogen bonds should be given to support this conclusion?
7. Please provide the absolute solid fluorescence quantum yield of compound 1 and the assembled system bearing host-guest
8. "No obvious wavelength shift was observed in the concentrated organic solutions (Supplementary Figures S23–27)." But, only S26-S27 showed that the compound 3 has no concentration effect in organic solutions. I suggest the authors double check every figure and all the data presented in both manuscript and SI.
9. There are many typographical and grammatical errors, which making the whole manuscript a bit unreadable.

Reviewer #1 (Remarks to the Author):

In this manuscript, Qu et al. designed a single fluorophore based on amphiphilic pyrene by introducing an acylhydrazone unit to broaden pyrene pi-conjugated backbone and to facilitate the intermolecular hydrogen-bonding interactions. On one hand, this single fluorophore could self-assemble into spherical particles in aqueous solution due to its inherent amphiphilic structure, thus leading to the formation of the excimer. On the other hand, gamma-cyclodextrin was used to draw pyrene fluorophores much closer for the red-shifted excimer emission. Moreover, the chemical input of alpha-amylase was utilized to decompose the macrocyclic gamma-CD, thus resulting in the programmable excimer emission. Interestingly, this multi-color fluorescent system could also be applied as self-erasing/ink materials. The conceptual design and methodology are novel and the conclusions are supported by the experimental results. Overall, in my view, the present work is suitable for publication in Nature Communications after addressing the following concerns.

A: We sincerely appreciate your positive comment and valuable suggestions, which facilitate the improvement of the manuscript. Accordingly, we have followed your suggestions and relevant updates have been added in the revised manuscript.

Comments:

1. *In Scheme 1, the cartoon diagram of host macrocycle in green should be marked as gamma-cyclodextrin and the corresponding chemical structure should be given.*

A: Thank you for your suggestions. The diagram of host macrocycle has been marked and the corresponding chemical structure has been given in the revised manuscript.

2. *In the part of "Amphiphilic self-assembly", the reduplicate TEM images in Figure 1C and S14 should be removed.*

A: Thank you for your suggestions. The reduplicate TEM images in both figure 1C and Supplementary material have been removed and replaced by the amplified TEM images in Figure 1C in the revised manuscript.

3. *In the part of "Amphiphilic self-assembly", is the morphology of the aqueous assemblies' spherical vesicles or micelles? It cannot be well assigned as bilayer structure from the existing representation. Is there any stronger evidence to prove that they are vesicles? Otherwise, amphiphilic nanoparticle may be an appropriate expression for the self-assembling morphology.*

A: Thank you for your valuable suggestions. The amplified TEM images (Figure 1C and Supplementary Figure S13) has been provided to prove the vesicles in the revised manuscript. The nanoparticles were hollow spheres with a diameter of about 110 nm and a membrane thickness of 2.6 ± 0.5 nm, suggesting the vesicle assemblies of compound **1** in water.

Figure 1C. Representative TEM images of compound **1** that assembles to form vesicles in an aqueous solution (25°C, 0.1 mM).

4. *The authors used the mixed solution of gamma-cyclodextrins and compound 1 to print the fluorescent images in Figures 3G and S37, but why was the color of the fluorescent patterning slightly different?*

A: We agree that this slight difference might cause unnecessary misunderstanding for readers. Hence, we removed the distorted photographs in the revised manuscript. The slight difference was the result of low concentration when the stored dye ink in the printer box almost run out.

5. *In the part of “Dissipative molecular emission system powered by chemical fuel”, the fluorescent color was unchanged even after 72 h in the absence of gamma-CDs, the picture should correspond to Supplementary Figures S39, not S30.*

A: Thanks for your careful reviewing, and the mentioned mistake has been corrected.

6. *Considering the importance of the 2:2 complex, more direct evidences were suggested to be provided to verify the formation of host-guest complex, such as 2D NOESY, and/or small-angle X-ray scattering (SAXS), and/or high-resolution mass spectrum, to comprehensively characterize the binding/aggregating mode between compound 1 and gamma-CD.*

A: Thank you for the valuable suggestions. We agree with your point that the 2:2 host-guest complex is important. Hence, following your suggestions, 2D NOESY and mass spectrum are performed to support the conclusion in the revised manuscript.

In the aqueous solutions, the proton signals of pyrene were shielded owing to the formation of the vesicle. After adding γ -CD (2 eq) to compound **1** aqueous solution (20 mM), the successful disassembly of vesicles and subsequent inclusion of pyrene units in γ -CD can be suggested by following observations: i) the appearance of aromatic proton signals suggests the decrease of shielding effect; ii) proton signals H_a , H_j and H_k split into two groups. To further verify the formation of 2:2 host-guest complex, 2D NOESY measurement was performed. The correlation signals of γ -CD with pyrene and pyrene with pyrene were observed. Notably, the

proton H₆ from the narrow rim of γ -CD was positioned close to proton H_k, suggesting that γ -CD is unidirectionally threaded onto the hydrophobic unit (*J. Am. Chem. Soc.* **2016**, 138, 13541-13550). In addition, the dimer of pyrene was confirmed by the correlation of the signals of (b, e), (i, e), (h, b), (d, j) and (g, k).

Figure S25. ¹H NMR (D₂O) spectra of compound **1** (20 mM) prior (top) and after (bottom) addition of 2 eq γ -CDs.

Figure S26. 2D NOESY NMR spectrum of the complex (D₂O, 25°C).

The 2:2 complex was further confirmed by the mass spectrum. Weak signals of the quaternary complexes were found at 2130.8, which consistent with the result: $[2M@2CD - 2H^+]^{2-}$ calcd for $C_{186}H_{274}N_4O_{106}^{2-}$, 2130.8126 or $[M@CD - H^+]$ calcd for $C_{93}H_{137}N_2O_{53}^-$, 2130.8126. Furthermore, the difference of m/z between isotope signals 2129.8, 2130.3 and 2130.8 were 0.5, indicated the presence of two charge states, i.e. $[2M@2CD - 2H^+]^{2-}$. This result strongly confirmed that compound **1** and γ -CD self-assembled into 2:2 complex in water.

Figure S27. Mass spectrum of the quaternary complexes.

7. *In Figure 1B, what does vertical coordinate (intensity/%) of DLS data refer to?*

A: Intensity/% refer to the intensity of laser light that was scattered by the particles moving at random by virtue of their Brownian motion. The particle size, number distributions and volume distributions were calculated by analysis of these intensity fluctuations. (Berne, B. J. & Pecora, R. Dynamic Light Scattering with Applications to Chemistry, Biology and Physics (Wiley, New York, **1976**).

8. *In the legend of Figure 1E, the value of CAC (90 μ M) is different from the description (9 μ M).*

A: The value of CAC in Figure 1E have been corrected in the revised manuscript.

9. *In Figure S13B, this reviewer cannot understand the large difference in length-width ratio. Why the height of spherical vesicle is 65 nm but the width is as large as 3 μ m?*

A: Thank you for the suggestions. The sample of AFM was prepared by the drop-casted of compound **1** aqueous solution on mica surface and followed by evaporation. The vesicles might collapse with the evaporation of the water hence the value of width was larger than height. Such phenomena also presented in other researches (*J. Am. Chem. Soc.* **2019**, 141, 6092-6107; *Angew. Chem. Int. Ed.* **2017**, 56, 5729-5733). Considering this result might cause unnecessary misunderstanding for readers. We consider that the amplified TEM images in Figure 3C have provided solid evidence and clear morphology for the assemblies, and the AFM image cannot represent the real morphology of vesicles in solution. Hence, we removed the AFM images from the revised manuscript.

10. *In Page 10, last line, the K_a value should be changed to $((6.11 \pm 0.48) \times 10^5 M^{-1})$.*

A: The mentioned issues have been corrected.

11. In page 11, last line, the pH value and ionic strength of PBS buffer should be given.

A: Thank you for your suggestion. The pH value of PBS buffer has been given in the revised manuscript. The calculation of the ionic strength is difficult, owing to the hydrolysis and ionization of the HPO_4^{2-} and H_2PO_4^- in solution. On the other hand, the ionic strength of buffer was rarely given in ITC data (*J. Am. Chem. Soc.* **2017**, 139, 3202-3208). Therefore, the ionic strength of PBS buffer was not given in the revised manuscript.

12. Authors claimed that the products of the enzymatic reaction resulted in the increase of lifetime and explained as feedback. This reviewer is curious about what kind of product decreased the reaction rate and how it occurred? Some reasonable explanations should be supplied.

A: Thank you for the valuable suggestions. Budapest et al., reported the hydrolysis of cyclodextrin by α -Amylase (*Starch* **1984**, 36, 140-143). Large quantities of maltooligomers (maltohexaose, maltopentaose, maltotetraose) were presented in the course of the hydrolysis. These oligomers are substrates of the enzyme, inhibiting the degradation of the cyclodextrin “competitively”. On the other hand, other products such as: glucose, maltose and maltotriose may link to the enzyme-protein, resulting in the “non-competitive” inhibitory effect.

Reviewer #2 (Remarks to the Author):

The manuscript by Qu et al. reports the color-tunable single-fluorophore, which showed interesting color-responding emission via assembled molecules. By using pyrene fluorophore and the acylhydrazone linker, the formed supramolecular system in aqueous media, gives the emissions ranging from blue to yellow, even a pure white light emission, while also shows fluorescent color differences within a slow hydrolysis reaction. In principle, there are certain interesting results in this contribution to smart fluorescent materials. However, there are several misleading descriptions and unclear presentations in the claim and the conclusion. This paper in the present form therefore could not be acceptable although significance and novelty of the content are high enough. A reconsider with major revisions should be recommended. Several important points to improve and strengthen the paper are listed below:

A: We appreciate your recognition on the novelty of this work, as well as many valuable suggestions. We have improved the manuscript by adapting your suggestions. We hope that this version is in good shape to be published.

1. The authors should be careful when the word “programmable”, do you think the emission is programmable? “precisely control the intermolecular aromatic stacking distances”, how to ‘precisely control’? these description and conclusion are overstated.

A: We agree with your point that these descriptions are slightly misleading. Hence, we have removed or replaced them in the revised manuscript.

2. Page 6. “.....transmission electron microscopy (TEM) (Supplementary Figures S13 and 14), showing spherical vesicles” please provide the amplified image of the vesicles in TEM, and give the Rg/Rh value of TEM.

A: Thank you for your valuable suggestion. The vesicles structure was well confirmed by the amplified TEM images (Figure 1C and Supplementary Figure S13). Compound **1** formed spherical particles with a diameter of about 110 nm and a membrane thickness of 2.6 ± 0.5 nm, suggesting the formation of vesicles in aqueous solution (*Angew. Chem. Int. Ed.* **2018**, *57*, 3132-3136).

Figure 1C. Representative TEM images of compound **1** that assembles to form vesicles in an aqueous solution (25°C, 0.1 mM).

3. *Page 7. “To confirm the formation of the excimer....., suggesting the presence of the excimer”. Please describe the excimer more clearly and explain in detail, and then give the conclusion.*

A: The appearance of a strong long-wavelength emission, with broad and unstructured characteristics, indicates the formation of excimers between pyrene units. As shown in Supplementary Figure S17, the excitation spectrum of this longer-wavelength component (530 nm) at higher concentration (20 μM , above CAC) was comparable to that of the monomer emission (460 nm) at lower concentration (5 μM , below CAC), and both of them resembled the absorbance spectra. These results further supported the 530 nm peak was indeed emission originating from the pyrene core, and this emission was consistent with an excited state dimerization process where, above CAC emission from the excimer dominates, whilst below CAC most excited state relaxation occurs prior to excimer formation. On this basis and supported by the photo-luminescence study, we attributed the 530 nm emission to an excimer species formed from a ground state and an excited pyrene unit induced by amphiphilic self-assembly.

4. *Page 7, “.....revealed a critical aggregation concentration (CAC) of 9 μM ” and “.....and versus the concentration revealed the CAC was 9 μM ”. While figure 1D and 1E which give 90 μM ?*

A: We are sorry for the mistake. The mentioned part has been corrected, and we also have checked the whole manuscript to eliminate similar mistakes.

5. *Page 8, “.....could be obtained in DMSO/H₂O mixtures with high water fractions (Supplementary Figure S20)” the data is not provided in supporting information “Figure S20”.*

A: The detailed data has been provided in Supplementary Figures S20.

6. *The authors claim “The potential intermolecular hydrogen bonds in the acylhydrazone units may work as secondary noncovalent interactions to.....”. In this case, evidences of hydrogen bonds should be given to support this conclusion?*

A: Thank you for the valuable suggestions. The intermolecular hydrogen bonds were confirmed by the FTIR spectra. The sharp amide I band shifted from 1658 cm^{-1} in DMSO (monomers) to 1645 cm^{-1} in H₂O (aggregation), suggesting the formation of intermolecular hydrogen bonds. These new results and demonstrations have been added into the revised manuscript.

Figure S16. FTIR spectra of compound **1** (10 mM) in DMSO (black line) and water (red line).

7. *Please provide the absolute solid fluorescence quantum yield of compound **1** and the assembled system bearing host-guest.*

A: Thank you for the valuable suggestions. Accordingly, we have added the new results as shown in Supplementary Figures S33 in the revised manuscript. Both compound **1** and the assembled system were measured three times and the average absolute solid fluorescence quantum yield were 17.05% and 47.99%, respectively.

Figure S33. The absolute solid fluorescence quantum yield of compound **1** (solid line) and the assembled system bearing host-guest (dash line).

8. *“No obvious wavelength shift was observed in the concentrated organic solutions (Supplementary Figures S23-27).” But, only S26-S27 showed that the compound **3** has no concentration effect in organic solutions. I suggest the authors double check every figure and all the data presented in both manuscript and SI.*

A: We are sorry for the mistake. The mentioned issues have been corrected and the figures as well as data presented in revised manuscript and SI have been double checked.

9. *There are many typographical and grammatical errors, which making the whole manuscript a bit unreadable.*

A: We are sorry for the errors in manuscript. Accordingly, we have significantly improved the English writing in the revised manuscript with the help of the technical writing editor in LetPub company.

REVIEWERS' COMMENTS:

Reviewer #1 (Remarks to the Author):

The authors have eliminated all my concerns in the revised manuscript, and this manuscript can be eventually accepted in Nature Communications now.

Just one minor revision left, it should be "13C NMR spectrum...", not "13C spectrum..." in the legend of Figure S11.

Reviewer #2 (Remarks to the Author):

In revised manuscript, several important points are well-clarified by the authors. Therefore, I would recommend the publication of this work in the Nature Communiacion.

Reviewer #1 (Remarks to the Author):

The authors have eliminated all my concerns in the revised manuscript, and this manuscript can be eventually accepted in Nature Communications now.

A: We sincerely appreciate your pertinent comments and valuable suggestions, which significantly improved our manuscript. The manuscript has been revised accordingly.

Comments:

Just one minor revision left, it should be “¹³C NMR spectrum...”, not “¹³C spectrum...” in the legend of Figure S11.

A: Thanks for your careful reviewing, and “¹³C spectrum” in the legend of Supplementary Figures 3, 7 and 11 have been corrected as “¹³C NMR spectrum”.

Reviewer #2 (Remarks to the Author):

In revised manuscript, several important points are well-clarified by the authors. Therefore, I would recommend the publication of this work in the Nature Communiacion.

A: We appreciate very much for your helpful suggestions and recommendation.